# Cervical cancer risk and access: Utilizing three statistical tools to assess Haitian women in South Florida

**Rhoda K. Moise** [1] *, **Raymond Balise**[2], **Camille Ragin**[3], **Erin Kobetz**[2]

**1** Patient Centered Care and Education, Research, Education, and Social Solutions, Inc. (REESSI), Hampton, Virginia, United States of America, **2** Department of Public Health Sciences, University of Miami Miller School of Medicine, Miami, Florida, United States of America, **3** Department of Cancer Prevention and Control, Fox Chase Cancer Center, Philadelphia, Pennsylvania, United States of America

* rmoise@reessi.com

**Data Availability Statement:** Data cannot be shared publicly because of confidentiality and there are legal and ethical restrictions being placed upon the data. Data contain potentially identifying or sensitive patient information and the University of

## Abstract

Although decreasing rates of cervical cancer in the U.S. are attributable to health policy, immigrant women, particularly Haitians, experience disproportionate disease burden related to delayed detection and treatment. However, risk prediction and dynamics of access remain largely underexplored and unresolved in this population. This study seeks to assess cervical cancer risk and access of unscreened Haitian women. Extracted and merged from two studies, this sample includes n = 346 at-risk Haitian women in South Florida, the largest U.S. enclave of Haitians (ages 30–65 and unscreened in the previous three years). Three approaches (logistic regression [LR]; classification and regression trees [CART]; and random forest [RF]) were employed to assess the association between screening history and sociodemographic variables. LR results indicated women who reported US citizenship (OR = 3.22, 95% CI = 1.52–6.84), access to routine care (OR = 2.11, 95%CI = 1.04–4.30), and spent more years in the US (OR = 1.01, 95%CI = 1.00–1.03) were significantly more likely to report previous screening. CART results returned an accuracy of 0.75 with a tree initially splitting on women who were not citizens, then on 43 or fewer years in the U.S., and without access to routine care. RF model identified U.S. years, citizenship, and access to routine care as variables of highest importance indicated by greatest mean decreases in Gini index. The model was .79 accurate (95% CI = 0.74–0.84). This multi-pronged analysis identifies previously undocumented barriers to health screening for Haitian women. Recent US immigrants without citizenship or perceived access to routine care may be at higher risk for disease due to barriers in accessing U.S. health-systems.

## Introduction

Worldwide, sexual and reproductive cancer, such as that of the cervix, remains a leading cause of death for women despite being preventable with early detection and treatment of cancerous lesions [1–5]. Although mortality from cervical cancer in the United States (U.S.) has significantly decreased over recent decades due to vaccination policies and screening recommendations, immigrants who are women of African ancestry still experience

Miami IRB has imposed them. Researchers can contact them through Cynthia Gates, JD, ADN, CIP, their Executive Director of Human Subjects Research Office (email:cmg345@med.miami.edu).

**Funding:** Authors thank and acknowledge Dinah Trevil, Tulay Koren-Sengul, Olveen Carrasquillo, and colleagues for sharing support in data acquisition. This work was supported by the National Institutes of Health (NIH) National Cancer Institute (NCI), 5-R01-CA-183612-02. Research reported in this publication was supported by the National Cancer Institute of the National Institutes of Health under Award Number P30CA240139. The content is solely the responsibility of the authors and does not necessarily represent the official views of the National Institutes of Health. The funders had no role in study design, data collection and analysis, decision to publish, or preparation of the manuscript. Publication was made possible in part with support from the GMaP Region 4 Research Stimulus Award.

**Competing interests:** The authors have declared that no competing interests exist.

disproportionate rates of disease [2, 4, 6–11]. Specifically, Haitian women have been found to have increased risk for cervical cancer in part due to delayed detection and subsequent treatment [4, 6–8, 12]. Yet, the literature to date under-investigates the risk factors that characterize health and health behavior for unscreened Haitian women living in the U.S.

Individuals of African ancestry may be African American or may belong to a multitude of other groups (e.g., French-, Dutch-, or English-speaking Caribbean/West Indian, African, American, European, Canadian). The African ancestry immigrant population has quadrupled since 1980 with over half of these immigrants originating from the Caribbean [13]. The Census Bureau projects that the number of immigrants of African ancestry will double again by 2060 with Haitian immigrants contributing a significant portion to the statistics [13]. Thus, Haitian women's health must be contextualized at the intersection of science, health, and policy, as well as through a syndemic, medical, and anthropological lens of structural violence [12, 14–18]. Structural violence may not be physical and refers to social institutions which harm groups of people by impeding their ability to achieve basic human needs such as healthcare, education, and other resources [19–21]. For instance, colonialism, sexism, racism, and xenophobia are so deeply embedded in society that they may go largely unnoticed while directly and indirectly impacting health and health behavior [20, 21].

Colonialism, sexism, racism, and xenophobia create a dynamic interplay of poverty, gendered experiences, and race consciousness, which must be exposed and rectified in order to promote health and prevent disease for all with efficiency and sustainability [12, 14–17]. Notably, xenophobia, the hatred of people from other countries, persists in the U.S. For instance, the CDC initially categorized four risks for HIV (referred to as 4H) including the following: hemophilia, heroin use, homosexuality, and people of Haitian ancestry [22, 23]. Although the CDC has redacted the wrongful statement regarding Haitian ancestry as a risk factor, Haitians still carry the burden of stigmatization and medical mistrust associated with the incident [15]. Haitians' unfavorable context of reception in the U.S. (e.g., government policies and local labor market, social relationships, and perceptions) predicts cultural dissonance, poor self-esteem, and depression which all consequently influence health and wellness [16, 17]. Thus, hostile context of reception may also provide insight into poor health outcomes in Haitian women [15–17]. Furthermore, Haitian women's native language is not English which may also cause challenges in navigating the U.S. health system. Overall, Haitian women are impacted by multiple structures of violence which may translate into restricted access to health preserving care like preventive gynecological measures against cervical cancer [12, 24, 25].

## Purpose of the study

This paper explores literature-supported variables including socio-demographic (age, marital and employment status, education), immigrant history (citizenship, length in U.S.), and health care utilization (insurance, access to routine care) to predict cervical cancer risk in Haitian women living in the U.S. [26]. Each variable has specific justification for inclusion in data analysis. For instance, socio-demographic variables are standard in statistics. Immigrant history adds contextual detail in data analysis for meaningful interpretation, and health care utilization links to health care outcomes [26]. We utilize three statistical methods: a) logistic regression (LR); b) classification and regression trees (CART); and c) random forest (RF) for comprehensive analytics. Statistical approaches vary in their degree of responding to scientific inquiry due to intrinsic assumptions for each methodology; therefore, this study includes three different methods to curate a comprehensive view of Haitian women's risk profiles for cervical cancer using predictors across immigration history, patterns of healthcare utilization, and key socio-demographic characteristics. Further, multiple statistical methodology allows for

examination of similarities and nuances across analysis, producing more reliable results and more detailed findings. Each statistical methodology holds inherent and applied (to this data-set) strengths and weaknesses. LR, RF, and CART models are all evaluated using a myriad of assessments, and this study uses accuracy, the systematic correctness of the model's predictions compared to the true classifications in the dataset where a high percentage is favored, as a common assessor of result comparison [27–29]. Further, through incorporating all three of the detailed statistical approaches, this study leverages strengths and minimizes weaknesses for a more comprehensive grasp of cervical cancer risk prediction [30].

## Theoretical rationale: PEN 3 Cultural Model's relationships and expectations via perceptions, enablers, and nurturers

The PEN-3 Cultural Model (Fig 1) has been developed particularly for research in ethnic populations [31–34]. Further, the PEN-3 Cultural Model has several domains which help researchers approach data analysis and interpretation with cultural sensitivity [31–34]. This study relates to the dimension of relationships and expectations (one's perceptions, enablers, and nurturers) by employing a series of quantitative analyses to assess the relationship between cervical cancer screening behaviors, immigration history, patterns of healthcare utilization, and key socio-demographic characteristics of Haitian immigrant women in the Miami metropolitan area (Fig 2). The PEN-3 Cultural Model includes categories to organize the data and aids in data interpretation to advance knowledge of socio-cultural influences of Haitian women's navigation of healthcare. Accordingly, we theoretically organized variables by perceptions, enablers, and nurturers of health and health behavior for the population [35]. For instance, participants' reports of age, access to routine care, and length of time in the U.S. make up the subdimension of perceptions. Enablers include participants' insurance and citizenship status. Education, employment, and marriage may nurture a participants' health and wellness.

Overall, this study aims to understand the characteristics of a Haitian woman living in South Florida who is at-risk for cervical cancer defined by her self-reported health history including absence of cervical cancer screening with a Pap test. Statistical findings may help to provide better comprehension of socio-cultural and socio-environmental factors fostering (or impeding) women immigrants' ability to navigate the U.S. healthcare system. Results may inform public health planning and delivery for Haitian women as well as other underserved immigrants living in the U.S. for equitable access to disease prevention and control.

## Materials and methods

### Sample

Data were extracted and merged from two larger studies, namely Health in Your Hands (HIYA; Clinical Trial Registration Number NCT02970136) and South Florida Center for the Reduction of Cancer Health Disparities (SUCCESS; Clinical Trial Registration Number NCT02121548). HIYA and SUCCESS tested modality, the fashion in which participants were screened (i.e., mailed self-sampler, community health worker [CHW] support). Both studies were ethically approved prior to the study by the University of Miami ethics committee as well as with an updated University of Miami IRB approval to be merged and analyzed (IRB #20180129). Both studies, HIYA and SUCCESS, included purposive convenience sampling. SUCCESS sought to bypass barriers to screening through delivering HPV self-sampling tools to women via a CHW [36]. Indeed, the study produced significant results indicating efficacy in screening uptake despite women's lack of knowledge, insurance, and access [37]. HIYA sought to clarify the viability of HPV self-sampling tools delivered by mail in order to circumvent

**Fig 1. The PEN-3 Cultural Model.** PEN-3 domain of relationships and expectations are the key serves as the theoretical framework element for this study.

conflict or confusion with scheduling an appointment with a CHW [4]. Findings from HIYA suggested no statistical differences in screening uptake by CHW compared to mailed kit, validating the feasibility of a new format for screening via a mailed self-sampling tool approach. Details of these studies may be found elsewhere [4, 36]. We also conducted preliminary data analysis to ensure there were no significant differences in the data to further justify merging. The sample was fairly split across both datasets ($n$ = 148 SUCCESS; $n$ = 198 HIYA) to include $n$ = 346 Haitian women living in South Florida who were at risk for cervical cancer, defined as self-reported age of 30–65 and unscreened in the previous three years.

**Fig 2. The PEN-3 Cultural Model relationships and expectations domain.** Relationships and expectations of the PEN-3 Cultural Model were applied to theoretically organize the study variables.

## Measures and analyses

Measures included self-reported information on previous Pap smear history, demographics, and health access. The response variable for all analyses was a binary, yes/no, outcome of whether women ever underwent a Pap smear. Study investigation focused on screening as compared to vaccination due to the demographics of the sample who immigrated to the U.S.

on average at an age beyond the recommended vaccination window of early adolescence. Explanatory measures included self-reported variables pertaining to immigration history, patterns of healthcare utilization, and socio-demographic characteristics. Measures were recoded into binary (yes/no) predictors. Immigration history included U.S. citizenship, where permanent resident status was coded as non-citizen. Healthcare utilization encompassed insurance and perceived access to routine medical care. Socio-demographic variables included marital status, education, and employment. For marital status, cohabitation, separated, divorced, and widowed were coded as not married. Education included less than high school or high school educated and above. Dataset (HIYA or SUCCESS) was also coded as binary. Although initially collected as continuous variables, age and length in U.S., both in years, were recoded categorically. Age was coded into terciles including 30–40 years of age or early-adulthood, 41–50 years of age or middle-adulthood, and 51–65 years of age or older-adulthood. Length of time spent in the U.S. was coded into quartiles including the following: = <5 years or recent immigrants, 6–25 years, 26–40 years, 41–50 years, and 51–65 years. All analyses were completed using R software version 3.5.1 [38]. All corresponding package numbers are included below.

**Three risk prediction processes across LR, CART, and RF.** To craft the most parsimonious model, data analysis included a manual hierarchal stepwise approach [39]. Each predictor was inputted for univariate logistic regression models to assess the association between screening history and various predictors. All predictors were fitted for a multivariate model to ascertain the variable inflation factor (VIF), a measure of violating the assumption of collinearity, and then predictors with a low VIF (<2.5) were then fitted for another multivariate model using the "step" function in R. All theory and literature-supported variables carried over from LR and were inputted for CART and RF. CART graphs were created for ease of interpretability with R packages including "rpart" version 4.1–13 and "rattle" version 5.2.0. The procedure of randomly creating 70:30 ratio for training (*n* = 241) to test (*n* = 105) sets, respectively, sufficed for considerations of model fit and accurate data representation. RF packages included "caret" version 6.0–80 with the default of *n* = 500 trees [40, 41].

## Results

### Sample characteristics

The sample included women who were on average 46 years of age (interquartile range = 39, 53; median = 46; SD = 9.2) with 30.5 years spent in the U.S. (interquartile range = 8, 47; median = 36; SD = 20.4). Women displayed low levels of citizenship (23.1%), insurance (19.4%), access to routine care (22.0%), and employment (28.3%). Slightly more than half of the sample was married (56.1%) with less than high school education (57.4%). The entire sample was foreign-born in Haiti, with the exception of three participants born in other parts of the Caribbean and Latin America. Although income data were collected, analyses were not viable as more than half of the sample lived below the poverty line (*n* = 190, 54.9%), with a sizeable portion of the sample reporting unknown income (*n* = 138, 39.9%). Overall the sample distribution presented as skewed with categorical binary variables representing a low socioeconomic status (SES) sample with recent immigration to the U.S. (<5 years), indicative of hyper-vulnerability to poor health outcomes [42]. The sample was fairly split across both datasets. Although none of the women adhered to U.S. recommendations of undergoing Pap smear screening every three years, classifying them as at risk for undetected cell abnormalities and potential infection, approximately two thirds of the sample (*n* = 226; 65.3%) had history of a previous Pap smear in general. Sample descriptive statistics are summarized in Tables 1 and 2. Results are described in detail below by statistical approach.

**Table 1. Sample descriptive statistics categorical variables (*n* = 346).**

| Variable | *n*(Percent) |
|---|---|
| Previous Pap | 226 (65.3) |
| U.S. Citizens | 80 (23.1) |
| Insured | 67 (19.4) |
| Routine Care | 76 (22.0) |
| Married | 194 (56.1) |
| Employed | 98 (28.3) |
| HS Educated | 152 (43.9) |
| Dataset (HIYA) | 148 (42.8) |

## LR, CART, and RF analytics

Results from both single predictor and multivariable logistic regression, predicting being screened, are depicted in Table 3. The models show increased odds of screening for women who were middle or older age, in the country for longer than five years, U.S. citizens, insured, employed, and able to access routine care. Marital status, education, and dataset did not produce significant univariate results.

A multivariate model was built including all variables to ascertain VIF. Two variables (years in the U.S. and dataset) were collinear, indicated by high VIF (>2.5). Given previous justification of merging across datasets as well as the insignificance of dataset in the univariate analyses, the dataset variable was consequently removed from the multivariate model, and categorical length in U.S. was maintained. In the next iteration of multivariate analyses, excluding the dataset variable, no predictors returned high VIF results. Further, middle and older age, mid-range length in the U.S., U.S. citizenship, and routine access to care were significant predictors of likelihood in presence of previous screening history. According to the C-statistic, 72% of the time this model would correctly guess, when presented with two women where only one of which was screened, which one was screened.

To further assess predictors, the R "step" function was employed to select the most parsimonious model. This required omitting 13 women from the categorical age and length in the U.S. predictors for model execution, leading to an *n* = 333 down from *n* = 346. The ultimate model shown in Table 4 was selected according to the PEN-3 Cultural Model and parsimony (AIC = 367.5) with age, citizenship, access to routine care, and education as predictors. Ultimately, women reporting middle or older age, U.S. citizenship, and access to routine care were most likely to report previous history of Pap smear screening. The C-statistic of 73% reflects a similar level of accuracy compared to the manually built model described above.

CART results, using the same dataset from initial analyses in LR with *n* = 346, returned an accuracy of 71.9% when predicting women's Pap smear history and a kappa of .37. The tree mainly split on women who were not citizens, in the U.S. for five years or fewer, and without routine access to medical care. Results of CART are depicted in graph form in Fig 3. Attempts to prune tree (c = .04) resulted with a notification that the fit was not a tree, just a root.

**Table 2. Sample descriptive statistics continuous variables (*n* = 346).**

| Variable | Average (Interquartile Range) |
|---|---|
| Age | 46 (39, 53) |
| Length in U.S. (years) | 30.5 (8, 47) |

**Table 3. Bivariate and multivariate logistic regression statistics.**

| Variables | | Bivariate OR, (95%CI) | Multivariate OR, (95%CI) |
|---|---|---|---|
| Age | | | |
| | A (30–40 years) | 1 (ref) | 1 (ref) |
| | B (41–50 years) | 2.10 (1.19, 3.72) * | 3.10 (1.19, 8.10) * |
| | C (51–65 years) | 2.29 (1.27, 4.11) ** | 2.56 (0.95, 6.89) |
| Length in U.S. | | | |
| | A (= <5 years) | 1 (ref) | 1 (ref) |
| | B (6–25 years) | 3.92 (1.81, 8.50) *** | 1.85 (0.76, 4.54) |
| | C (26–40 years) | 2.45 (1.23, 4.86) * | 2.89 (1.14, 7.35) * |
| | D (41–50 years) | 3.19 (1.59, 6.38) ** | 1.52 (0.64, 3.63) |
| | E (51–65 years) | 3.83 (1.79, 8.19) *** | 1.72 (0.63, 4.67) |
| U.S. Citizenship | | 4.94 (2.44, 10.00) *** | 3.19 (1.36, 7.49) ** |
| Insured | | 2.09 (1.12, 3.90) * | 1.12 (0.51, 2.46) |
| Routine Care | | 3.18 (1.67, 6.06) *** | 2.60 (1.16, 5.81) * |
| Married | | 1.01 (0.65, 1.59) | 0.94 (0.55, 1.59) |
| Employed | | 2.09 (1.23, 3.57) ** | 1.06 (0.55, 2.02) |
| HS Educated | | 1.10 (0.07, 1.74) | 1.58 (0.89, 2.82) |
| Dataset (HIYA) | | 1.41 (0.90, 2.21) | N/A |

Logistic regression statistics for the bivariate and multivariate analyses.

Note: Significance codes: 0 '***' 0.001 '**' 0.01 '*' 0.05 '.' 0.1 ' ' 1.

OR = Odds ratio; CI = confidence interval; Ref = Referent group; N/A = Not applicable.

RF results, also using the same dataset from analyses in LR and CART with $n = 346$, identified years spent in the U.S., citizenship, and age as variables of highest importance indicated by greatest mean decreases in the Gini index. A loop function to identify best settings confirmed the defaults were ideally maintained as 500 trees and mtry three for a 37.6% error. The training model was 78% accurate (95% CI = 0.72–0.83) with specificity of 0.94, the predictive ability to correctly identify women who had not been screened, and a Positive Predictive Value (PPV) of 82%. The testing model was 72% accurate (95% CI = 0.61–0.81) with specificity of 0.80, the predictive ability to correctly identify women who had not been screened, and a Positive Predictive Value (PPV) of 50%. Table 5 contains statistical results for the RF confusion matrices of the training and test sets. Fig 4 displays variables of importance accordingly.

**Table 4. Stepwise regression model ($n = 333$).**

| Variables | | Stepwise OR, (95%CI) |
|---|---|---|
| Age | | |
| | A (30–40 years) | 1 (ref) |
| | B (41–50 years) | 2.21 (1.17, 4.17) * |
| | C (51–65 years) | 1.96 (0.98, 3.94) |
| U.S. Citizenship | | 4.11 (1.83, 9.26) *** |
| Routine Care | | 2.69 (1.26, 5.71) * |
| HS Educated | | 1.60 (0.91, 2.80) |

Stepwise regression statistics for the multivariate analysis.

Note: Significance codes: 0 '***' 0.001 '**' 0.01 '*' 0.05 '.' 0.1 ' ' 1.

OR = Odds ratio; CI = confidence interval; Ref = Referent group; N/A = Not applicable.

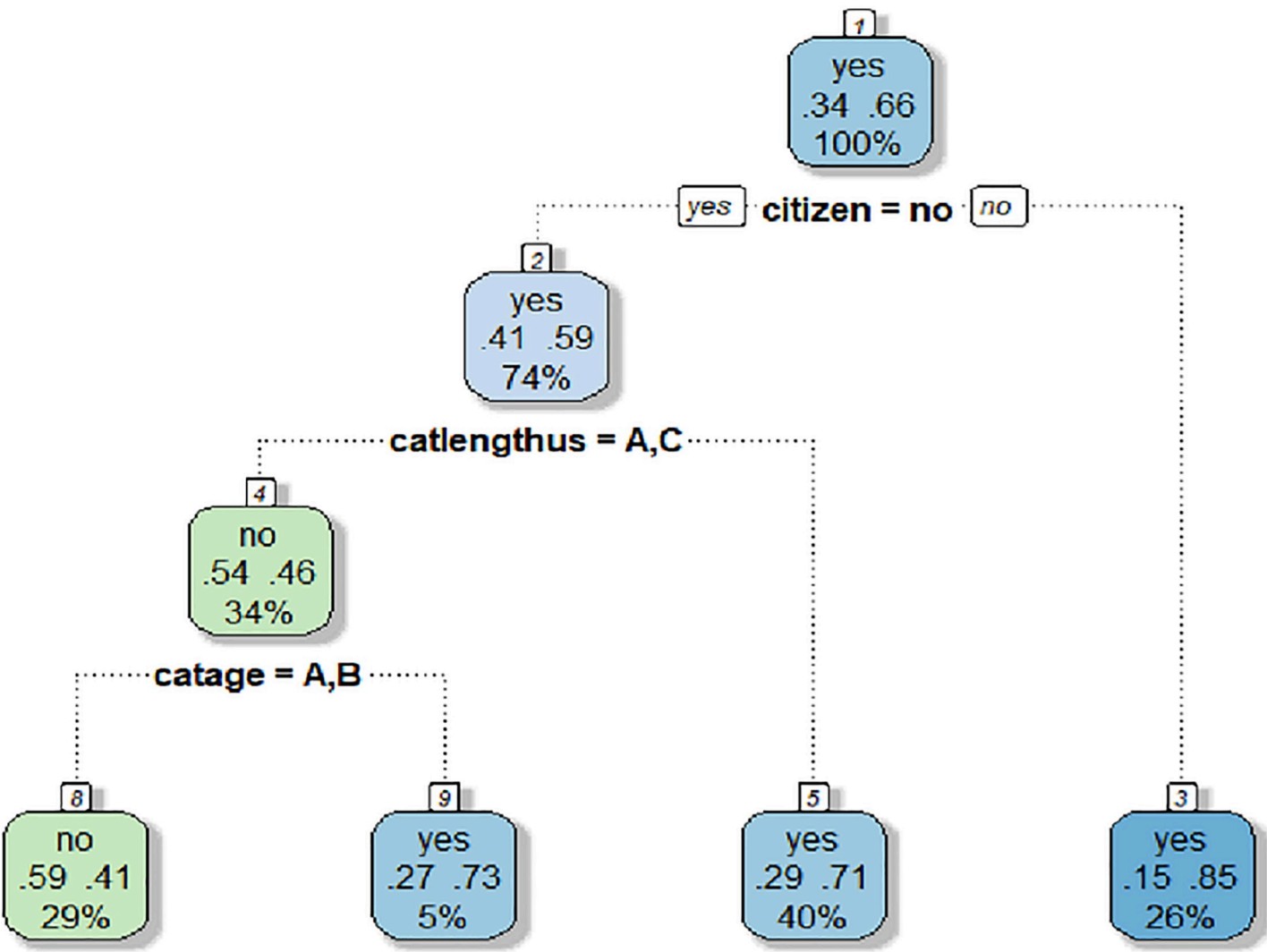

**Fig 3. Decision tree–CART model, fancy rpart plot.** There are numeric and text outputs associated with respective splits to aid in result interpretation. The figure featured at the top represents the predicted value per split. The number featured at the lower left represents the likelihood of the predicted split value. The number featured at the lower right represents the likelihood of the opposite endorsement value. The percentage at the bottom represents the size of the sample passing through the node per split. For example, with node number 2, the group is inclined to take on the activity, "yes" regarding screening history. The probability for people to say "yes" in this group is .41 while the probability for the group to take on the opposite activity, "no," is .59. Of the 346 study subjects, 74% of the data passed through this node. Overall, the decision tree selected citizenship, length of time in the U.S., and age as important predictors in ascertaining a women's screening history. For example, with node 1, the model asks if the participant is not a citizen. If the participant responds affirmatively, she is not a citizen, she will be classified into the node on the left. If the participant responds in opposition, "no" they are indeed as citizen, she will be classified in the node on the right.

## Discussion

This study utilizes three quantitative methods (LR, CART, and RF) for improved assessment of risk of cancer for Haitian women who have never undergone screening, based in Miami-Dade County, FL, the largest enclave of Haitians living in the U.S. [37]. This study uniquely applies classic analysis of LR, an innovative application of CART, and rigorous assessment of RF, to understand cervical cancer risk in Haitian women. This study utilized three statistical methods to explore risk prediction for Haitian women who had never undergone screening for cervical cancer from a sample of women who reported no screening in the previous three years. Results indicated numerous factors may influence the disproportionate burden of HPV

**Table 5. Confusion matrix statistics for random forest (RF) training (*n* = 241) and test sets (*n* = 105).**

| Statistic Type | Training Set Result | Test Set |
|---|---|---|
| Accuracy | 0.78 | 0.72 |
| 95% CI | (0.72, 0.83) | (0.61, 0.81) |
| No Information Rate | 0.65 | 0.72 |
| P-Value [Acc > NIR] | 3.01e-05 | 0.55 |
| Pos Pred Value | 0.82 | 0.50 |
| Neg Pred Value | 0.77 | 0.81 |
| Specificity | 0.94 | 0.80 |
| Sensitivity | 0.47 | 0.52 |
| McNemar's Test P-Value | 4.84e-06 | 1.00 |
| Kappa | 0.46 | 0.31 |
| Prevalence | 0.35 | 0.28 |
| Detection Prevalence | 0.20 | 0.29 |
| Balanced Accuracy | 0.71 | 0.66 |

infection and related cancer in Haitian women including lack of screening, non-citizenship status, recent immigration to the U.S., and routine access to care.

Reports of non-citizenship were the strongest predictors of absence of screening history across all analytic tools followed by younger (<40 years of age), recent immigrants (<5 years). LR results suggest perceived access to routine care outperforms insurance in predictive power which underscores the interplay of multilevel factors influencing access to health care for immigrant populations [4, 37, 43, 44]. However, uncertainty remains regarding the importance of other socio-demographic variables. For instance, education, employment, access to care, marital status, and insurance did not significantly appear in CART yet were present in varying degrees of LR analyses as well as the output of important variables produced by the random forest. RF models were specific, yet not very sensitive, meaning the model functioned better in predicting those under-screened than those who were screened. This finding is meaningful and useful due to the study's purpose of identifying women at risk [41]. Although no cutoff was provided for the length in the U.S. predictor, it was the most important variable. Such findings demonstrate the need to further tease out levels of risk by immigration time [24, 45–47].

## PEN-3 Cultural Model–relationships and expectations via perceptions, enablers, and nurturers

Framed by the PEN-3 Cultural Model, results indicate enablers as the most influential portion of the model considering citizenship status appearing as a key predictive variable across all analytics while insurance had little effect. Next, perceptions followed in significance with length of time in the U.S. and access to care. Finally, nurturing variables (theorized as education, employment, and marriage) had mild influence in predicting a participant's screening history. These findings add complexity to the literature on the healthy migrant effect which suggests a selection bias of groups in better health tend to immigrate [42, 48, 49]. Although the theory may hold true for baseline health, it may not accurately describe health *behaviors*. Future studies should consider qualitatively exploring drivers of HPV screening knowledge, attitudes, and behaviors (KAB) as well as access and barriers to improve Haitian women's uptake [44, 50].

## Strengths and limitations

This study should be considered in light of both strengths and weaknesses. For instance, this study includes vigorous statistical methodology and robust data analysis [27]. Additionally, the

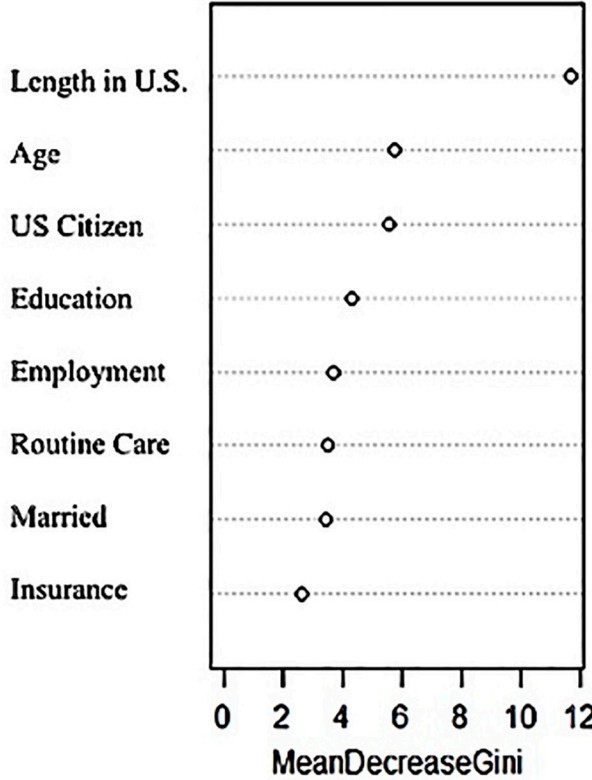

**Fig 4. Random Forest (RF) important variables.** Results of RF variables of importance are featured including order of ranked importance.

sample size was relatively small for the type of analyses typically used for machine learning tools [51–55]. The sample also included a skewed sample of Haitian women with low SES indicated by a majority of below poverty or unknown income. In spite of the limited, medium sample size, with low SES Haitian women, the consistency of results along with their associated data validation metrics producing accuracy across all methods are encouraging.

## Conclusions

Although some recognize systemic, structural violence as a problem and are working to address it, women still encounter related difficulties and barriers to accessing care such as sexual and reproductive health [20, 56]. Particularly, Black women experience hyper-vulnerability to inequity and societal exclusion at the intersection of sexism and racism [57, 58]. Health and research policies and procedures have been institutionalized in order to account for such biases and prevent poor practices in the future. These practices included, for examples, federal entities requiring assessments of inclusion of women and minorities in grants, institutional review boards, and informed consent. However, detrimental effects on vulnerable populations still carry over from historical context [59].

Further, findings imply the need to address related systematic barriers blocking Haitian women from successfully navigating the U.S. healthcare system due to colonialism, sexism, racism, and xenophobia [32, 60]. Citizenship status may supersede literature supported variables in driving health and health behavior [12, 26, 46]. Overall, results mainly highlighted younger, recent women immigrants, without citizenship, as the most at risk for cervical cancer

due to lack of screening. There may also be implications for compounded factors and cumulative effects of other variables such as employment, education, and marital status. Future studies must consider holistic ecological approaches to population health to best serve Haitian women spanning their individual insights (i.e., KAB) to institutional influences (i.e., SES, citizenship). Implications for health equity and public health entail policy and systems level consideration for vulnerable populations in light of structural violence linked to gender, racial/ethnic and, nativity.

## Author Contributions

**Conceptualization:** Rhoda K. Moise, Raymond Balise, Erin Kobetz.

**Data curation:** Rhoda K. Moise, Raymond Balise, Erin Kobetz.

**Formal analysis:** Rhoda K. Moise, Raymond Balise.

**Funding acquisition:** Erin Kobetz.

**Investigation:** Rhoda K. Moise, Erin Kobetz.

**Methodology:** Rhoda K. Moise, Raymond Balise, Camille Ragin.

**Project administration:** Erin Kobetz.

**Resources:** Rhoda K. Moise, Camille Ragin.

**Software:** Rhoda K. Moise, Raymond Balise.

**Supervision:** Rhoda K. Moise, Raymond Balise, Erin Kobetz.

**Validation:** Rhoda K. Moise, Raymond Balise.

**Visualization:** Rhoda K. Moise.

**Writing – original draft:** Rhoda K. Moise, Erin Kobetz.

**Writing – review & editing:** Rhoda K. Moise, Raymond Balise, Camille Ragin, Erin Kobetz.

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
