## [Decision Letter · Decision Letter 0]

12 Jan 2021

PONE-D-20-21793

Cervical Cancer Risk and Access: Utilizing Three Statistical Tools to Assess Haitian Women in South Florida

PLOS ONE

Dear Dr. Moise,

Thank you for submitting your manuscript to PLOS ONE. After careful consideration, we feel that it has merit but does not fully meet PLOS ONE’s publication criteria as it currently stands. Therefore, we invite you to submit a revised version of the manuscript that addresses the points raised during the review process.

I would like to sincerely apologise for the delay you have incurred with your submission. It has been exceptionally difficult to secure reviewers to evaluate your study. We have now received two completed reviews; their comments are available below. Please revise the manuscript to address all the reviewer's comments in a point-by-point response in order to ensure it is meeting the journal's publication criteria. Please note that the revised manuscript will need to undergo further review, we thus cannot at this point anticipate the outcome of the evaluation process.

We look forward to receiving your revised manuscript.

Kind regards,

Miquel Vall-llosera Camps

Senior Editor

PLOS ONE

Journal Requirements:

2. Please provide more information on the data sources used. We note that two different clinical trials are cited as  data sources ; please ensure that you provide more information on the two trials (settings,  participants included, etc) and discuss what type of data were used in your study (baseline/ post-intervention). Moreover, please clarify the eligibility criteria used for inclusion in your analysis; and provide a participant flowchart.

3. Please specify whether you had access to any identifying information, and provide the IRB approval number for your study.

4. We suggest you thoroughly copyedit your manuscript for language usage, spelling, and grammar. If you do not know anyone who can help you do this, you may wish to consider employing a professional scientific editing service.  

6. Please respond by return e-mail with an updated version of your manuscript to include your abstract after the title page.

7. Please include a separate caption for each figure in your manuscript.

**Comments to the Author**

1. Is the manuscript technically sound, and do the data support the conclusions?

Reviewer #1: Yes

Reviewer #2: Yes

2. Has the statistical analysis been performed appropriately and rigorously? 

Reviewer #1: Yes

Reviewer #2: Yes

3. Have the authors made all data underlying the findings in their manuscript fully available?

Reviewer #1: No

Reviewer #2: Yes

4. Is the manuscript presented in an intelligible fashion and written in standard English?

Reviewer #1: Yes

Reviewer #2: No

5. Review Comments to the Author

Reviewer #1: This study examined factors associated with cervical cancer screening among Haitian women. Specifically, the authors used three different statistical methods to determine salient factors related to lack of screening among. I have offered a few suggestions for the authors’ consideration:

Abstract

•The authors state, “This study seeks to assess cancer risk and access of unscreened Haitian women.” The statement seems to indicate an examination of general cancer risk as opposed to risk and access specific to cervical cancer.

Introduction

•The first sentence of the second paragraph is missing a closed parenthesis. Same issue for the last sentence of paragraph three.

•The description of the influence of how structural violence influence health behaviors is an interesting concept. However, since the study focuses on specific determinants of cancer screening, which have been examined in previous immigrant health literature, it would be helpful to provide further justification for choosing the specific screening determinants in this study.

•The authors indicate that the study utilizes three distinct statistical methods to examine factors associated with Pap screening. It seems to me that the focus of the paper is on the use of the statistical methods, so it would be helpful to provide additional justification for using this approach.

Methods

• The theoretical rationale seems a bit out of place in the methods sections. I suggest including this in the introduction section. Perhaps, it may bolster the arguments for examining factors associated with Pap screening using three statistical methods.

• Why were data from the two studies merged, I assume both data were very similar? What was the sample sizes for each data?

Results

• In subsection 3.2 (LRC, CART, and…) include the relevant statics, e.g. the ORs and CIs.

Discussion

• The first paragraph of the discussion provides more detailed rationale for utilizing the three statistical methods in this paper. I suggest incorporating some of the information here in the introduction section.

Tables

• In table 1, I suggest including information for the categorical variables. For instance, what is the breakdown of length in the US, education, age etc.?

Reviewer #2: Introduction:

Your problem specification needs to be clearer, for example in page 8, you mentioned “structural violence, colonialism, racism and xenophobia” which are note in your aim of study.

You should specify the reference for each sentence in your introduction, for example in page 9 the sentence:” Colonialism, sexism, racism, and xenophobia create a dynamic interplay of poverty, gendered experiences, and race consciousness, which must be exposed and rectified in order to promote health and prevent disease for all with efficiency and sustainability.” Has not any references. There are many such sentences that you need to add references for them.

At the end of the introduction, please mention the aim of your study.

Methods:

Please explain why you chose PEN 3-Cultural Model for your study.

In page 10, part 2.1 the last sentence should be transferred to the introduction.

Please also mention the criteria and reason you considered for selecting each predictor.

Since the number of your data is low, the results are not reliable, so please try applying cross validation to gain more reliable results.

Results:

The accuracy obtained from the approaches may be improved, try selecting more important variables for your prediction and report them if the results improved.

6. PLOS authors have the option to publish the peer review history of their article (what does this mean?). If published, this will include your full peer review and any attached files.

Reviewer #1: No

Reviewer #2: **Yes: **Farkhondeh Asadi

---

## [Author Response · Author response to Decision Letter 0]

24 Mar 2021

Thank you for the feedback. I responded to all comments in an uploaded attached file.

---

## [Decision Letter · Decision Letter 1]

14 May 2021

PONE-D-20-21793R1

Cervical Cancer Risk and Access: Utilizing Three Statistical Tools to Assess Haitian Women in South Florida

PLOS ONE

Dear Dr. Moise,

Thank you for submitting your manuscript to PLOS ONE. After careful consideration, we feel that it has merit but does not fully meet PLOS ONE’s publication criteria as it currently stands. Therefore, we invite you to submit a revised version of the manuscript that addresses the points raised during the review process.

We look forward to receiving your revised manuscript.

Kind regards,

Farkhondeh Asadi

Academic Editor

PLOS ONE

Journal Requirements:

Additional Editor Comments (if provided):

Dear Authors,

Thanks for addressing some of reviewers' comment. According to reviews' 1 comment he/she believes that his/her comment did not address well so, I invite you to response to his/her comment again correctly.

Reviewers' comments:

Reviewer's Responses to Questions

**Comments to the Author**

1. If the authors have adequately addressed your comments raised in a previous round of review and you feel that this manuscript is now acceptable for publication, you may indicate that here to bypass the “Comments to the Author” section, enter your conflict of interest statement in the “Confidential to Editor” section, and submit your "Accept" recommendation.

Reviewer #1: (No Response)

Reviewer #2: All comments have been addressed

2. Is the manuscript technically sound, and do the data support the conclusions?

Reviewer #1: Yes

Reviewer #2: Yes

3. Has the statistical analysis been performed appropriately and rigorously? 

Reviewer #1: Yes

Reviewer #2: Yes

4. Have the authors made all data underlying the findings in their manuscript fully available?

Reviewer #1: (No Response)

Reviewer #2: Yes

5. Is the manuscript presented in an intelligible fashion and written in standard English?

Reviewer #1: Yes

Reviewer #2: Yes

6. Review Comments to the Author

Reviewer #1: (No Response)

Reviewer #2: Dear authors, Thank you very much for improving your manuscript, all the comments are well addressed.

7. PLOS authors have the option to publish the peer review history of their article (what does this mean?). If published, this will include your full peer review and any attached files.

Reviewer #1: No

Reviewer #2: **Yes: **Farkhondeh Asadi

---

## [Author Response · Author response to Decision Letter 1]

21 May 2021

May 18, 2021

PLOS ONE

Re: Manuscript Submission PONE-D-20-21793

Dear Dr. Miquel Vall-llosera Camps,

My colleagues and I would like to thank you for allowing us to revise and resubmit our manuscript, “Cervical Cancer Risk and Access: Utilizing Three Statistical Tools to Assess Haitian Women in South Florida”. My colleagues and I feel that the comments and concerned raised by you and the reviewers have significantly strengthened the manuscript. 

There was confusion regarding the submission process and Response to Reviewers letter. I have resubmitted our response letter (please see the end of this PDF) where we list each comment made by each of the reviewers, followed by a narrative response to the comment. We have also included requested editorial revisions. We have track changes for ease of reference as well as a clean, updated document. As previously mentioned, Dr. Elizabeth Metzger, a professor in the English department at University of South Florida, provided copyediting services. In order to incorporate responses to the many excellent comments of the reviewers, the manuscript is over length. If there are additional comments from Reviewer 1 who felt their feedback was not incorporated, please have them specify their concerns. Please communicate the confusion to them to ensure they note that their comments were considered, but due to confusion where I included responses to reviewers previously, then got an admin request for updates, the original track changed document and responses were not sent to them. I valued their insights.

I also maintain the following by kindly offering more clarification in response to your previous request to make the data publicly available. The data includes a vulnerable group facing stigma. Further, the IRB did not include approval of data sharing. Thus, we are ethically unable to grant public availability of the data. However, there may be potential opportunities to collaborate with interested researchers. The data findings of the larger studies are available in published in peer-reviewed journals as indicated in the manuscript. The corresponding author’s information is available for contact accordingly.

I hope this addresses the data issue. I noted in my previous resubmission that "Data cannot be shared publicly because of confidentiality. Data are available from the Dr. Erin Kobetz at University of Miami (contact via ekobetz@med.miami.edu) for researchers who meet the criteria for access to confidential data. Indeed, there are legal and ethical restrictions being placed upon the data. Data contain potentially identifying or sensitive patient information and the University of Miami IRB has imposed them. Feel free to contact them through Cynthia Gates, JD, ADN, CIP, their Executive Director of Human Subjects Research Office (email:cmg345@med.miami.edu)"

Please let me know if you need anything else from me in order to move forward. Thank you in advance for considering our manuscript for publication. Overall, we hope that you and the reviewers will find our revised manuscript acceptable for publication in PLOS ONE. We have been corresponding for quite some time with minor updates. Please note, this paper is time sensitive as my access to funds to pay for the publication will expire in July. Should you have any questions or concerns, please feel free to contact me. 

Sincerely,

Rhoda Moise, Ph.D. rmoise@reessi.com

---

## [Decision Letter · Decision Letter 2]

21 Jun 2021

Cervical Cancer Risk and Access: Utilizing Three Statistical Tools to Assess Haitian Women in South Florida

PONE-D-20-21793R2

Dear Dr. Moise,

We’re pleased to inform you that your manuscript has been judged scientifically suitable for publication and will be formally accepted for publication once it meets all outstanding technical requirements.

Kind regards,

Farkhondeh Asadi

Guest Editor

PLOS ONE

Additional Editor Comments (optional):

Reviewers' comments:

Reviewer's Responses to Questions

**Comments to the Author**

1. If the authors have adequately addressed your comments raised in a previous round of review and you feel that this manuscript is now acceptable for publication, you may indicate that here to bypass the “Comments to the Author” section, enter your conflict of interest statement in the “Confidential to Editor” section, and submit your "Accept" recommendation.

Reviewer #1: All comments have been addressed

2. Is the manuscript technically sound, and do the data support the conclusions?

Reviewer #1: Yes

3. Has the statistical analysis been performed appropriately and rigorously? 

Reviewer #1: Yes

4. Have the authors made all data underlying the findings in their manuscript fully available?

Reviewer #1: No

5. Is the manuscript presented in an intelligible fashion and written in standard English?

Reviewer #1: Yes

6. Review Comments to the Author

Reviewer #1: (No Response)

7. PLOS authors have the option to publish the peer review history of their article (what does this mean?). If published, this will include your full peer review and any attached files.

Reviewer #1: No

---

## [Editor Report · Acceptance letter]

24 Jun 2021

PONE-D-20-21793R2 

Cervical cancer risk and access: Utilizing three statistical tools to assess Haitian women in South Florida 

Dear Dr. Moise:

I'm pleased to inform you that your manuscript has been deemed suitable for publication in PLOS ONE. Congratulations! Your manuscript is now with our production department. 

Kind regards, 

on behalf of

Dr. Farkhondeh Asadi 

Guest Editor

PLOS ONE